# High-accuracy spinal alignment monitoring using the head angle and visual distance in computer users

Ko Hashimoto[1]*, Yusuke Sekiguchi[2], Kaho Matsuda[3], Masataka Hori[2,4], Yutaka Mizuno[2,5], Ryosuke Shibuya[2], Kohei Takahashi[1], Takahiro Onoki[1], Kenichiro Yahata[1], Shin-Ichi Izumi[2,6], Toshimi Aizawa[1]

1 Department of Orthopaedic Surgery, Tohoku University Graduate School of Medicine, Sendai, Miyagi, Japan, 2 Department of Rehabilitation Medicine, Tohoku University Graduate School of Medicine, Sendai, Miyagi, Japan, 3 Non-Profit Organization "natural science", Sendai, Miyagi, Japan, 4 Department of Physical Therapy, Faculty of Health and Medical Sciences, Iryo Sosei University, Iwaki, Fukushima, Japan, 5 Department of Physical Therapy, Sendai Health and Welfare Professional Training College, Sendai, Miyagi, Japan, 6 Tsurumaki-Onsen Hospital, Hadano, Kanagawa, Japan

* ko.hashimoto.b1@tohoku.ac.jp

## Abstract

### Study Design

A Prospective Validation Study.

### Objectives

To validate a novel, noninvasive method for estimating the spinal sagittal alignment during seated computer work, using the head angle (HA) and visual distance (VD) as primary parameters.

### Methods

A 3D motion analysis system measured HA and VD in 21 healthy volunteers. The relationship between these parameters and spinal sagittal alignment, as determined by body surface markers, was investigated. To validate this method, radiographic measurements were taken in a separate group of 32 patients to confirm the link between body surface landmarks and actual spinal alignment. Additional variables, including gender, age, height, and weight, were incorporated into the model to improve accuracy.

### Results

HA and VD showed significant correlations with spinal sagittal alignment, particularly for the cervical spine (C2-C7). Incorporating demographic factors further enhanced the predictive accuracy. Radiological validation confirmed that body surface marker-based measurements are closely aligned with standard radiographic indices widely used in spine surgery.

**Data availability statement:** All relevant data are within the manuscript and its Supporting Information files.

**Funding:** The authors received no specific funding for this work.

**Competing interests:** The authors have declared that no competing interests exist.

## Conclusions

This study introduces a reliable and practical method for continuously monitoring spinal sagittal alignment in seated computer users. The approach demonstrates high accuracy, particularly for the cervical spine and holds promise for the development of posture-monitoring technologies to help prevent neck and back pain associated with prolonged computer use.

## Introduction

During and after the COVID-19 pandemic, the amount of time people work from home and look at computers and electronic devices has increased dramatically [1]. Remote work during the pandemic has been associated with higher incidence of musculoskeletal pain [2], and negative effects on work productivity [3]. Maintaining proper sagittal spinal alignment, the shape of the spine in an anteroposterior direction, is a key factor in preventing spinal issues during daily work. Neck stiffness and computer vision syndrome are known to result from prolonged computer use in a seated position [4]. Real-time information on spinal alignment in everyday life could help individuals recognize poor posture, maintain good posture, and better understand the cause-and-effect relationship between spinal malalignment and associated symptoms. Traditionally, spinal alignment monitoring has been conducted using radiographic imaging [5] or body surface scanning methods such as Moiré topography for screening [6]. These approaches provide reliable and reproducible spinal alignment data, particularly for routine spinal deformity check-ups. However, they require expensive and bulky equipment, making them impractical for continuous monitoring in daily life. To date, continuous and quantitative sagittal alignment monitoring has not been available. We propose utilizing head angle (HA) relative to the horizontal plane and the visual distance (VD) between the eyes and the computer display, -two parameters that can be continuously tracked using a small, wearable device (such as smart glasses) (Fig 1). If this estimation method proves reliable, spinal sagittal alignment could be continuously monitored and reported to individuals during daily computer use. The aim of this study is to validate the feasibility of estimating spinal sagittal alignment in a seated position using HA and VD.

## Materials and methods

### Method overview and expected outcome

The validation experiments for estimating spinal sagittal alignment using HA and VD consisted of i) three-dimensional (3D) motion analysis and ii) measurement of spinal sagittal alignment using whole-spine lateral X-ray, utilizing a different set of subjects. In the 3D motion analysis, sagittal spinal alignment obtained from the body surface markers (objective variables) was used to derive a quadratic linear approximation (estimated formula) through multiple regression analysis with HA, VD, gender, age, height, and weight as explanatory variables, the correlation coefficient and the coefficient of determination were calculated. For radiographic measurement, correlation coefficients were

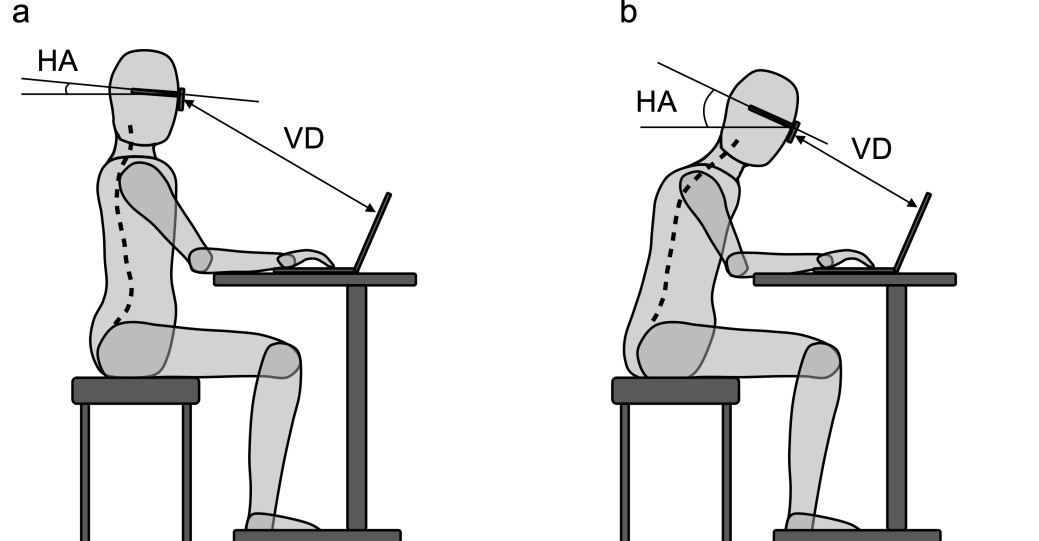

**Fig 1. Figures depicting the estimation of the spinal sagittal alignment using the head angle (HA) and visual distance (VD) as explanatory parameters.** Broken lines represent the spinal sagittal alignment. (a) The spinal column is well-aligned and upright with a smaller HA and longer VD. (b) The spinal column leans forward with a larger HA and shorter VD.

determined between sagittal alignment values obtained from the body surface indices as used in the 3D motion analysis and actual sagittal spinal parameters widely used in spine surgery. This radiographic validation confirmed that the sagittal alignment measurements based on body surface indices were sufficiently correlated with the actual sagittal spinal parameters. The intra- and inter-observer agreement was also evaluated for radiographic measurement. Ideally, these validations would be conducted on the same subjects. However, for ethical reasons unnecessary radiographic imaging of healthy volunteer subjects was not feasible. Therefore, in this research, healthy volunteers were used for 3D motion analysis, while patients who had already undergone whole-spine lateral radiographs in an outpatient clinic were used for radiographic validation.

## Subjects

Three-dimensional (3D) motion analysis was performed on 21 healthy volunteers. Participants were recruited from personal networks between November 30, 2020, and September 30, 2021. Efforts were made to ensure diversity in physical characteristics, such as height and body weight, as well as an equal distribution of male and female participants. The demographic data of the examinees are shown in S1 Table. The inclusion criteria required participants to have no history of spinal surgery. A power analysis conducted by a professional statistical expert was used to estimate the sample size necessary for constructing a predictive model based on three-dimensional motion analysis data. Assuming an expected coefficient of determination ($R^2$) of 0.4 and a mean-to-standard deviation ratio of 1.0, the analysis indicated that a minimum of 10 samples was required, with each contributing more than 10,000 data points.

To validate spinal sagittal alignment measurement by body surface landmarks on whole spine lateral X-ray, 32 independent patients from our outpatient clinic for bone metabolism treatment were randomly selected. The patients aged 37–54 years, were included without specific inclusion or exclusion criteria. The demographic data of the examinees are shown in S2 Table. A power analysis conducted by a professional statistical expert determined the sample size needed to evaluate the correlation ($R = 0.5$) between actual spinal sagittal parameters and sagittal alignment measured using body surface landmarks in X-rays. Assuming a significance level ($\alpha$) of 0.05 and a statistical power of 0.8, the analysis indicated that at least 29 samples were required. To account for potential variability, the sample size was increased to 32.

### Three-dimensional (3D) motion analysis (Fig 2A)

The subjects wore glasses while seated with motion capture markers placed on the glasses (two markers on the right frame for HA and VD measurements), on the body surface at the spinous processes of C2, C7, T3, T8, T12, L3 and S1, and at the top-center and bottom-center on the display of a laptop computer (for VD measurement). The vertebrae were identified by a spine surgeon (M.D., Ph.D.) with > 20 years' experience in spinal anatomy, while the subject was seated. The method was similar to that described in previous studies [7,8].To accurately and precisely identify each spinal level, the spinous processes were palpated and counted from the second cervical vertebra (C2) downward and from the first sacral vertebra (S1) upward. At the beginning of the experiment, the subject was instructed to sit in a chair (height: 40 cm) in front of a table (height: 70 cm) with a laptop computer. The subject was instructed to lean forward as much as possible while maintaining focus on the screen. The examinee followed the slow upward and downward movement of a ball displayed on the screen, coordinating neck movement, accordingly after each up and down movement until the examinee slightly tilted the trunk backward until reaching the limit of backward trunk tilting. Eight infrared cameras (Raptor-H, Raptor Photonics, Larne, Northern Ireland, UK) were used for three-dimensional motion analysis. These cameras were positioned at a height of 2.4 meters, evenly spaced in a circular arrangement surrounding the subject. Calibration was performed

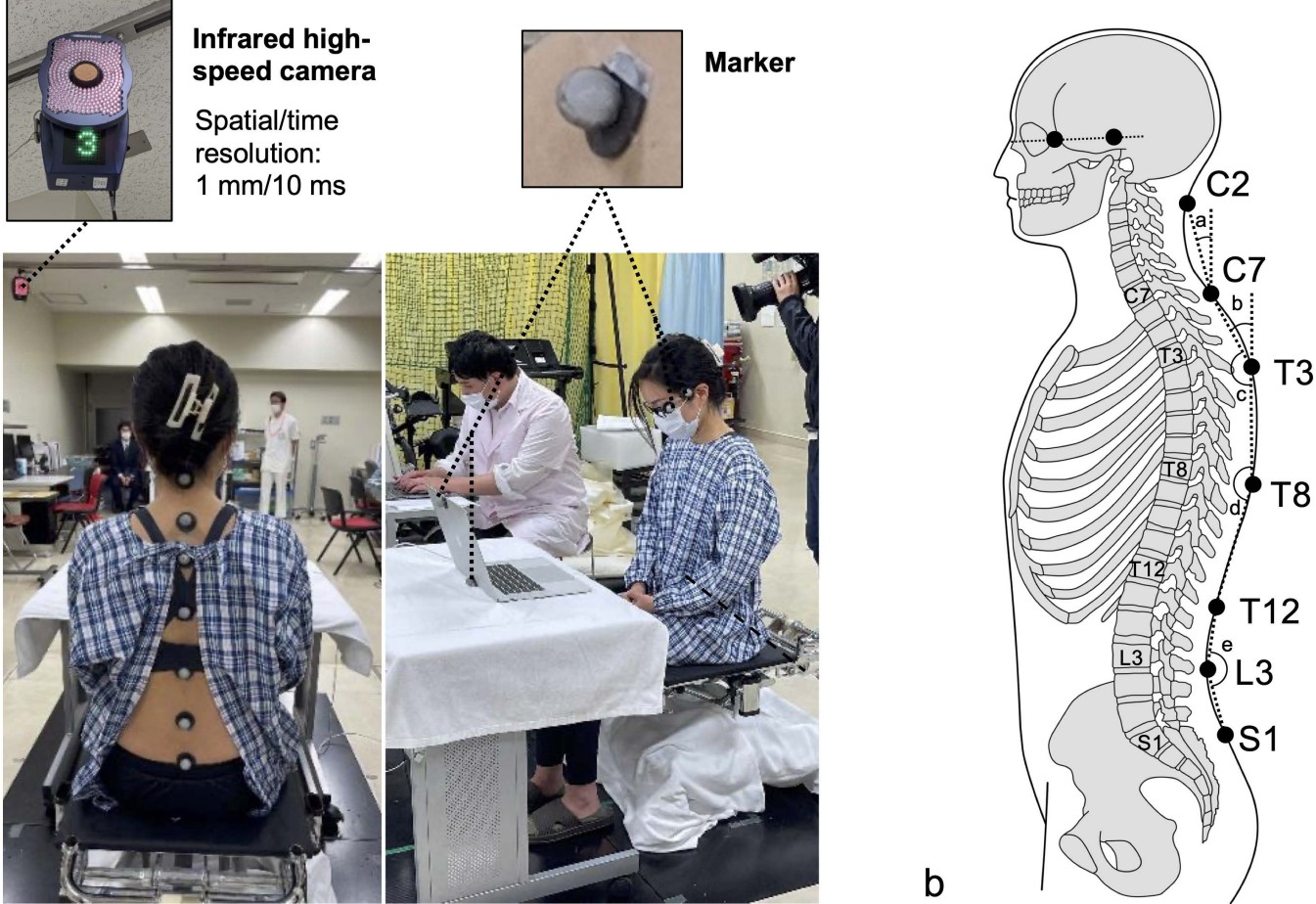

**Fig 2. Three-dimensional (3D) motion analysis experiment with marker placement on the body surface, resulting in spinal sagittal parameters.**
(a) Marker placement and 3D motion analysis scene. (b) A Schematic representation of marker placement and generated for spinal sagittal alignment. a: C2-C7 tilt angle, b: C7-T3 tilt angle, c: C7-T3-T8 angle, d: T3-T8-T12 angle and e: T12-L3-S1 angle.

according to the workstation protocol (MAC 3D, Motion Analysis Corporation, Santa Rosa, CA). Specifically, during validation experiments, a rod with two markers placed at 50-cm intervals was moved around various locations within the marker detection range. This process was repeated until a signal indicating sufficient marker detection accuracy was achieved. The 3D coordinates of each marker were recorded at 100 Hz (per second) at a resolution of 1 mm. To improve the reliability of the subsequent correlation assessment, each subject repeated movements in the same manner three consecutive times without repositioning the body-mounted markers to generate a greater number of datasets.

### Evaluated parameters

From the 3D coordinate data obtained from the motion analysis, the sagittal component coordinates were extracted and used for the angle and distance analysis of the sagittal plane. To remove outliers from the continuous time-series data (collected at 0.01-second intervals), a moving average was calculated using a window of 100 consecutive data points (equivalent to 1 second), centered on each data point. Data points that deviated by more than 100% from the moving average were considered outliers and removed. This criterion was verified because, during slow and continuous human movements, it is unlikely that marker coordinates will change by more than 100% within a 1-s period. The investigations were conducted using a combination of Microsoft Excel software (Redmond, WA), GraphPad Prism 9 software (Dotmatics, San Diego, CA) and an original C++-based program. The HA was defined as the angle formed between the horizontal line and the line segment connecting the two markers attached to the temples of the glasses. The VD was defined as the distance between the front-side marker the vane of the glasses and the midpoint of the two markers on the laptop computer. Based on the cleaned coordinate data, we calculated the inclination of the line segments connecting adjacent markers placed on the spinous processes relative to the vertical axis, as well as the angles formed between these adjacent segments. Each evaluated angle represents or simulates an angle commonly used for X-ray evaluation of sagittal spinal alignment in the field of spine care. The detailed information of the evaluated angles is shown in Fig 2B and Table 1.

### Multiple regression analysis

i)  Conduct individual analyses for each examinee in each trial.

First, each data set obtained from 3 trials by 21 examinees was analyzed independently (a total of 63 data sets: with approximately 10,000 data points per trial). The correlation between HA/VD and each of the above angles was investigated by multiple regression analysis and represented by a quadratic linear approximation equation, with HA and VD as explanatory variables as shown below.

**Table 1. Evaluated angles by 3D-coordinates obtained in 3D-motion analysis.**

| Parameter | Explanation |
| --- | --- |
| C2-C7 tilt angle ("a" in Fig 2) | Angle made by a line connecting markers at C2-C7 and a vertical line representing anterior tilting of the cervical spine |
| C7-T3 tilt angle ("b" in Fig 2) | Angle made by a line connecting markers at C7-T3 and a vertical line representing anterior tilting of upper thoracic spine |
| C7-T3-T8 angle ("c" in Fig 2) | Angle made by lines connecting markers at C7-T3 and T3-T8, simulating upper thoracic kyphosis |
| T3-T8-T12 angle ("d" in Fig 2) | Angle made by lines connecting markers at T3-T8 and T8-T12, simulating middle to lower thoracic kyphosis |
| T12-L3-S1 angle ("e" in Fig 2) | Angle made by lines connecting markers at T12-L3 and L3-S1, simulating lumbar lordosis |

3D: three dimensional.

$$\text{e.g., (C2 - C7 tile angle)} = b_1(HA)^2 + b_2(VD)^2 + b_3(HA) + b_4(VD) + b_5$$

bx: constant

The correlation coefficient (R) and coefficient of determination ($R^2$) were the Pearson product-moment correlation coefficients that were then computed for each combination. The scattered plots were created to show trends in the correlation coefficient (R) and the coefficient of determination ($R^2$) between HA/VD and each angle (objective variable).

ii) Analysis combining all the data sets by the 21 examinees

Next, all the data sets obtained from three trials involving 21 examinees were combined for analysis (approximately 630,000 data points total). The correlation between HA/VD and each of the above angles was analyzed as same as i), with the calculation of "R" and "$R^2$" in the same way. Furthermore, sex (male = 1, female = 0), age (years), height (cm), or weight (kg) of the examinees was separately enrolled as an explanatory variable in addition to HA/VD, as shown below.

$$\text{e.g., (C2 - C7 tile angle)} = b_1(HA) + b_2(HA)^2 + b_3(VD) + b_4(VD)^2 + b_5(Age) + b_6(Age)^2 + b_7$$

$b_x$: constant

Finally, all four parameters were included in multiple regression analysis as shown below with the same "R" and "$R^2$" calculations.

$$\text{e.g., (C2 - C7 tile angle)} = b_1(HA) + b_2(HA)^2 + b_3(VD) + b_4(VD)^2 + b_5(Sex) + b_6(Age) + b_7(Age)^2$$
$$+ b_8(Height) + b_9(Height)^2 + b_{10}(Weight) + b_{11}(Weight)^2 + b_{12}$$

$b_x$: constant

**Validating spinal sagittal alignment measurement by body surface landmarks using whole-spine lateral X-ray**

A retrospective analysis of lateral whole-spine radiographs was conducted on 32 independent outpatients (3 males, 29 females) who were not participants in the 3D motion analysis study. Spinal angles were measured using the same anatomical landmarks across the spinous processes of C2, C7, T3, T8, T12, L3, and S1. Lateral whole-spine radiographs were obtained while the subject (patient) stood upright with both upper limbs naturally relaxed at their sides. A long-cassette film capable of capturing the entire spine in a single exposure was used at a tube-to-film distance of 150 cm. Table 3 shows the subjects' demographic data. In addition, the commonly used radiographic parameters to assess the sagittal alignment of the spine, such as the center of gravity (COG)-C7 sagittal vertical axis (SVA), C2-C7 SVA, cervical lordosis (Cobb C2-C7), T1 slope, thoracic kyphosis (T1-T8 and T5-T12) and lumbar lordosis, were measured in each patient. Fig 3 and Table 2 show the measurement methods, and the meaning of the parameters used to assess the spine's sagittal alignment. The measurement was conducted by two examiners, twice by K.H. and once by K.T. The examiners conducted the measurements independently in a randomized order, with both the subjects' information and each other's results blinded. The same examiner took two measurements separated by a two-week interval. To validate the measurement of spinal sagittal alignment by body surface landmarks using lateral whole-spine radiographs, a simple linear regression was performed to analyze the correlation between local spinal sagittal alignment measured by body surface landmarks (e.g., C2-C7 anterior tile angle) and spinal sagittal parameters measured on radiographs (e.g., C2-C7 SVA), combining data from three measurements by two examiners, as previously stated. Each analysis included a calculation of the correlation coefficient (R). The data were analyzed using GraphPad Prism 9 software (Dotmatics, San Diego, CA), to evaluate intra- and inter-observer agreement, intraclass correlation coefficients (ICC) were calculated for the variables used in the above analyses

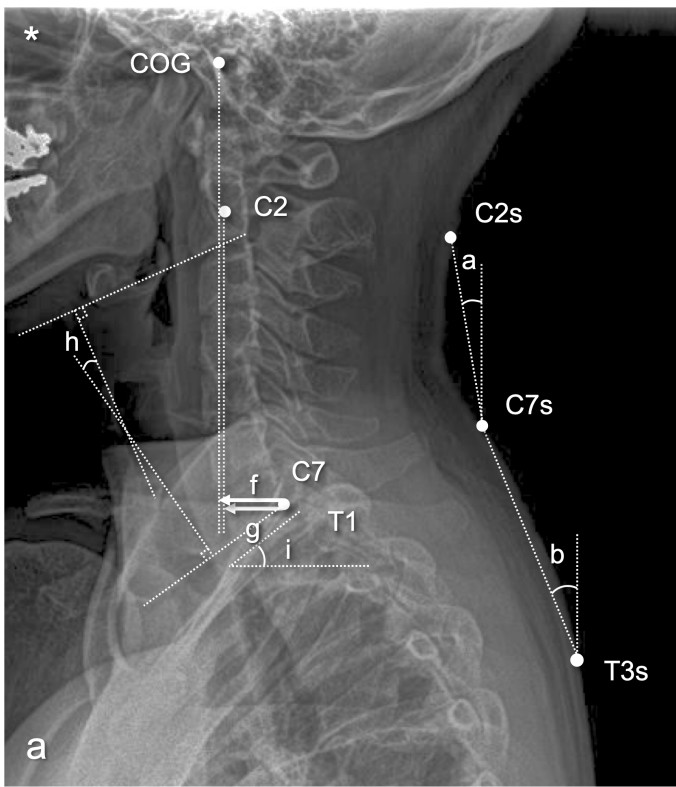
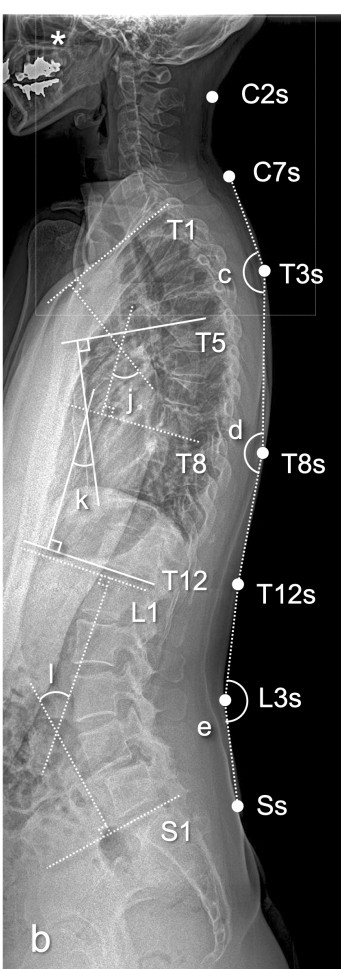

**Fig 3. Spinal sagittal alignment can be measured radiographically using a lateral X-ray of the entire spine.** (a) Magnified image of the boxed area in the whole-spine lateral radiograph on the right. (b) A whole-spine lateral radiograph. COG: center of gravity, Spinal level with "s" (e.g., C2s): indicates the spinal level's body surface point.

of radiographic measurements using Microsoft Excel software (Redmond, WA). Intraobserver agreement (ICC(1,1)) was determined using measurement data from two reviews by K.H. Interobserver agreement (ICC(2,1)) was calculated using data from K.T. and two reviews by K.H. According to previous evaluation criteria for ICC [9], ICC values of 0–0.2, 0.2–0.4, 0.4–0.6, 0.6–0.8, and 0.8–1.0 were classified as slight, fair, moderate, substantial, and almost perfect agreement, respectively. An independent statistician reviewed and validated the statistical analysis methods and results.

The study was conducted in accordance with the Declaration of Helsinki (as revised in 2013). The study was approved and individual consent for this analysis was waived by the Institutional Ethics Board of the authors' institute (Approval Number: 2020-1-368). All the participants appearing in this study provided informed consent for the publication in an open access journal.

## Results

### Three-dimensional motion analysis

i) Analyze each examinee for each trial (see S1 Fig)

**Table 2. Definition of radiographic parameters of spinal sagittal alignment commonly used in the field of spine surgery.**

| Parameter | Explanation |
|---|---|
| COG-C7 SVA ("f" in Fig 3) | Distance between plumb lines dropped from anterior margin of external auditory meatus (COG) and posterior superior corner of C7 |
| C2-C7 SVA ("g" in Fig 3) | Distance between a plumb line dropped from the centroid of C2 and the posterior superior corner of C7 |
| Cervical lordosis ("h" in Fig 3) | Sagittal Cobb angle between upper endplate of C2 and lower endplate of C7 (positive in lordotic angle) |
| T1 slope ("i" in Fig 3) | Angle made between upper endplate of T1 and horizontal line |
| Upper thoacic kyphosis ("j" in Fig 3) | Sagittal Cobb angle between upper endplate of T1 and lower endplate of T8 (positive in kyphotic angle) |
| Lower thoracic kyphosis ("k" in Fig 3) | Sagittal Cobb angle between upper endplate of T5 and lower endplate of T12 (positive in kyphotic angle) |
| Lumbar lordosis ("l" in Fig 3) | Sagittal Cobb angle between upper endplate of L1 and upper endplate of S1 (positive in lordotic angle) |

COG: center of gravity, SVA: sagittal vertical axis.

In a total of 63 trials by 21 subjects, the correlation coefficients (R) between HA/VD and the C2-C7 tilt angle, C7-T3 tilt angle, C7-T3-T8 angle, T3-T8-T12 angle and T12-L3-S1 angle were 0.9939 (0.9905–0.9952), 0.9957 (0.9930–0.9976), 0.9174 (0.8046–0.9672), 0.7347 (0.5904–0.8559) and 0.8689 (0.7346–0.9320), respectively (median; bracket: 25–75 percentile). The coefficients of determination ($R^2$) were 0.9879 (0.9811–0.9905), 0.9914 (0.9861–0.9953), 0.8416 (0.6474–0.9355), 0.5398 (0.3486–0.7325) and 0.7550 (0.5396–0.8686), respectively (median; bracket: 25–75 percentiles). The scatter plots of R and $R^2$ are shown in S1 Fig.

ii) Analysis of all data sets from 21 examinees (Figs 4 and 5, and Table 3 and 4)

C2-C7 tilt angle and C7-T3 tilt angles were correlated with HA, VD, and HA/VD by an "R" of 0.7086, 0.8332 and 0.8861, and 0.5641, 0.8377 and 0.8460, respectively, in a combined analysis of 630,000 data points from all trials by 21 examinees. Age and gender were the only explanatory variables that increased the "R" of the C2-C7 tilt angle and the C7-T3 tilt angle the most from 0.8861 to 0.9389 and from 0.8460 to 0.8885, respectively. After combining all the additional explanatory parameters in addition to HA/VD, the "R" of the C2-C7 tilt angle and the C7-T3 tilt angle increased to 0.9479 and 0.9021, respectively. Conversely, for the thoracic and lumbar spine, the C7-T3-T8, T3-T8-T12, and T12-L3-S1 angles had lower correlations with HA/VD, with all values less than 0.3.

The "$R^2$" between the C2-C7 tilt angle and HA/VD and between the C7-T3 tilt angle and HA/VD were 0.7852 and 0.7157, respectively. Meanwhile, for the thoracic and lumbar spine, the $R^2$ values for the C7-T3-T8, T3-T8-T12, and T12-L3-S1 angles to HA/VD were less than 0.1, even after incorporating all the additional explanatory parameters. Figs 4 and 5 and Tables 3 and 4 present detailed data. The coefficients on each explanatory variable in the C2-C7 tilt angle and C7-T3 tilt angle estimation formulas are also listed in S4 and S5 Tables, respectively.

iii) Correlation between the spinal alignment measured by body surface landmarks and sagittal spinal radiographic parameters on X-ray (S2 Fig)

In the radiographic analysis, the R between the C2-C7 tilt angle and COG-C7 SVA, the C2-C7 tilt angle and C2-C7 SVA, the C2-C7 tilt angle and CL, the C7-T3 tilt angle and the T1 slope, the T3-T8-T12 angle and the T1-T8 kyphosis, the T3-T8-T12 angle and the T5-T12 kyphosis, and the T12-L3-S1 angle and LL were 0.7259, 0.7775, −0.1810, 0.7700, 0.5670, 0.8304 and 0.6185, respectively. All the above combinations except for the C2-C7 tilt angle and CL were significantly correlated, with a P-value of less than 0.0001. The $R^2$ exceeded 0.5 in combinations between the C2-C7 tilt angle

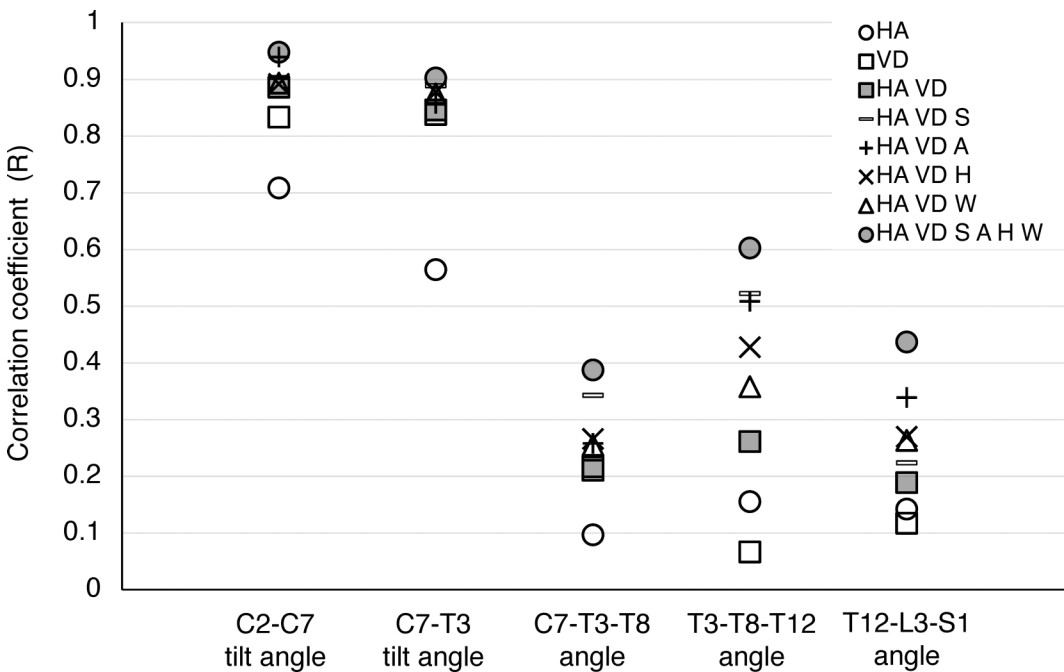

**Fig 4. Correlation coefficient (R) between sagittal spinal angles obtained by 3D-motion capture analysis and incorporated explanatory parameters.** HA: head angle, VD: visual distance, S: sex, A: age, H: height, W: weight.

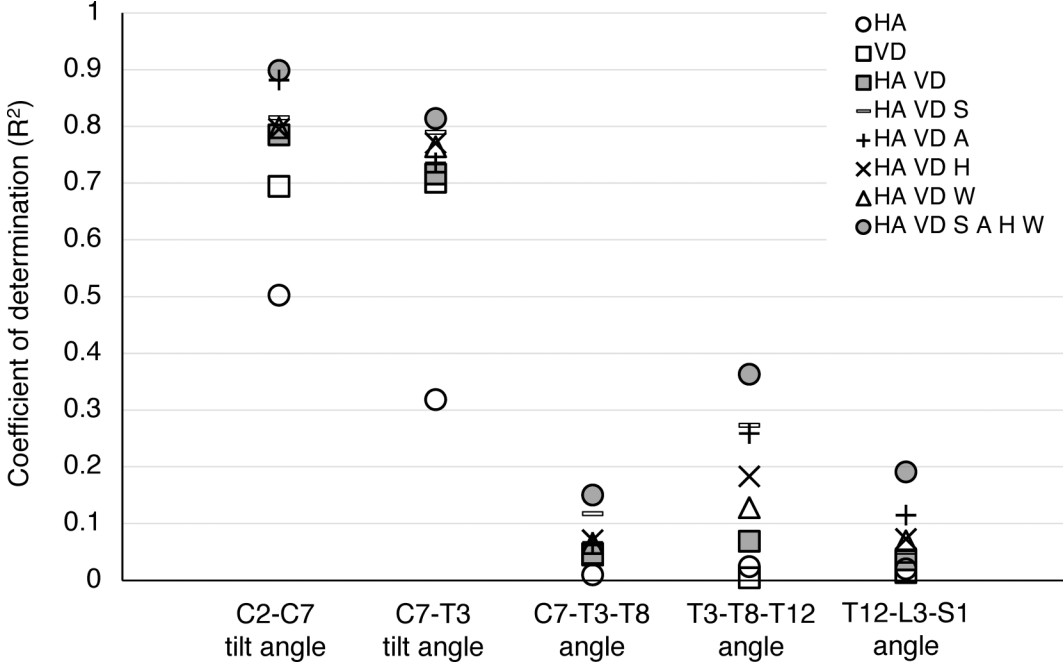

**Fig 5. Coefficient of determination ($R^2$) between sagittal spinal angles obtained by 3D-motion capture analysis and incorporated explanatory parameters.** HA: head angle, VD: visual distance, S: sex, A: age, H: height, W: weight.

**Table 3. Correlation coefficient (R) between sagittal spinal angles obtained by 3D-motion capture analysis and incorporated explanatory parameters.**

| Enrolled explanatory parameters | Objective parameters | | | | |
|---|---|---|---|---|---|
| | C2-C7 tilt angle | C7-T3 tilt angle | C7-T3-T8 angle | T3-T8-T12 angle | T12-L3-S1 angle |
| HA | 0.7086 (0.7079 - 0.7092) | 0.5641 (0.5635 - 0.5648) | 0.0967 (0.0965 - 0.0969) | 0.1551 (0.1549 - 0.1553) | 0.1424 (0.1422 - 0.1426) |
| VD | 0.8332 (0.8330 - 0.8334) | 0.8377 (0.8376 - 0.8379) | 0.2104 (0.2103 - 0.2105) | 0.0667 (0.0666 - 0.0669) | 0.1174 (0.1172 - 0.1176) |
| HA VD | 0.8861 (0.8860 - 0.8862) | 0.8460 (0.8458 - 0.8462) | 0.2165 (0.2164 - 0.2166) | 0.2615 (0.2613 - 0.2618) | 0.1887 (0.1885 - 0.1888) |
| HA VD S | 0.9030 (0.9029 - 0.9031) | 0.8885 (0.8884 - 0.8887) | 0.3426 (0.3424 - 0.3428) | 0.5225 (0.5222 - 0.5227) | 0.2235 (0.2233 - 0.2237) |
| HA VD A | 0.9389 (0.9388 - 0.9389) | 0.8576 (0.8574 - 0.8578) | 0.2579 (0.2577 - 0.2580) | 0.5085 (0.5082 - 0.5086) | 0.3391 (0.3389 - 0.3393) |
| HA VD H | 0.8918 (0.8917 - 0.8919) | 0.8776 (0.8775 - 0.8778) | 0.2656 (0.2654 - 0.2657) | 0.4278 (0.4276 - 0.4281) | 0.2701 (0.2699 - 0.2703) |
| HA VD W | 0.8928 (0.8927 - 0.8929) | 0.8739 (0.8737 - 0.8741) | 0.2547 (0.2545 - 0.2549) | 0.3579 (0.3577 - 0.3581) | 0.2633 (0.2631 - 0.2635) |
| HA VD S A H W | 0.9479 (0.9478 - 0.9480) | 0.9021 (0.9019 - 0.9023) | 0.3878 (0.3875 - 0.3881) | 0.6024 (0.6021 - 0.6027) | 0.4366 (0.4364 - 0.4368) |

HA: head angle, VD: visual distance, S: sex, A: age, H: height, W: weight.

Values in brackets indicate 95% confidence interval.

**Table 4. Coefficient of determination ($R^2$) between sagittal spinal angles obtained by 3D-motion capture analysis and incorporated explanatory parameters.**

| Enrolled explanatory parameters | Objective parameters | | | | |
|---|---|---|---|---|---|
| | C2-C7 tilt angle | C7-T3 tilt angle | C7-T3-T8 angle | T3-T8-T12 angle | T12-L3-S1 angle |
| HA | 0.5021 (0.5003 - 0.5038) | 0.3183 (0.3165 - 0.3200) | 0.0094 (0.0092 - 0.0095) | 0.0241 (0.0238 - 0.0243) | 0.0203 (0.0201 - 0.0205) |
| VD | 0.6942 (0.6929 - 0.6954) | 0.7018 (0.7005 - 0.7030) | 0.0443 (0.0441 - 0.0444) | 0.0045 (0.0043 - 0.0046) | 0.0138 (0.0136 - 0.0140) |
| HA VD | 0.7852 (0.7842 - 0.7861) | 0.7157 (0.7143 - 0.7171) | 0.0469 (0.0467 - 0.0471) | 0.0684 (0.0681 - 0.0687) | 0.0356 (0.0354 - 0.0358) |
| HA VD S | 0.8154 (0.8146 - 0.8162) | 0.7895 (0.7883 - 0.7907) | 0.1174 (0.1169 - 0.1178) | 0.2730 (0.2724 - 0.2736) | 0.0500 (0.0497 - 0.0502) |
| HA VD A | 0.8814 (0.8809 - 0.8820) | 0.7355 (0.7340 - 0.7370) | 0.0665 (0.0663 - 0.0667) | 0.2585 (0.2580 - 0.2590) | 0.1150 (0.1146 - 0.1154) |
| HA VD H | 0.7953 (0.7943 - 0.7963) | 0.7702 (0.7689 - 0.7715) | 0.0706 (0.0703 - 0.0708) | 0.1830 (0.1827 - 0.1834) | 0.0730 (0.0727 - 0.0732) |
| HA VD W | 0.7972 (0.7966 - 0.7977) | 0.7637 (0.7624 - 0.7651) | 0.0649 (0.0646 - 0.0651) | 0.1281 (0.1278 - 0.1284) | 0.0693 (0.0691 - 0.0696) |
| HA VD S A H W | 0.8985 (0.8980 - 0.8990) | 0.8138 (0.8126 - 0.8150) | 0.1504 (0.1499 - 0.1509) | 0.3629 (0.3623 - 0.3630) | 0.1906 (0.1902 - 0.1910) |

HA: head angle, VD: visual distance, S: sex, A: age, H: height, W: weight.

Values in brackets indicate 95% confidence interval.

and COG-C7 SVA, the C2-C7 tilt angle and C2-C7 SVA, the C7-T3 tilt angle and the T1 slope and the T3-T8-T12 angle and the T5-T12 kyphosis. The detailed data are shown in Table 5.

iv) ICC values for intraobserver and interobserver agreement (S3 Table)

The intraobserver agreement of the body surface parameters ranged from 0.5612 to 0.9232, indicating moderate-to-almost perfect agreement based on the previous evaluation criteria ICCs [9]. The intraobserver agreement of the radiographic parameters ranged from 0.8172 to 0.9721, indicating nearly perfect agreement in all parameters assessed. Conversely, the interobserver agreement of the body surface parameters ranged from 0.5573 to 0.9046, indicating moderate-to-almost perfect agreement. The interobserver agreement of the radiographic parameters ranged from 0.6556 to 0.9844, indicating substantial to almost perfect agreement. The detailed data are shown in S3 Table.

## Discussion

In the current era of prolonged visual display terminal work, poor seated posture is linked to neck and/or lower back pain [10–12]. Continuous monitoring of the sagittal alignment of the spine, i.e., spinal shape in the anteroposterior direction, has the potential to help individuals become more aware of their poor posture and avoid neck/lower back pain. Meanwhile, assessing the spinal sagittal alignment has traditionally necessitated the use of radiographic equipment, which is not readily available for monitoring an individual's daily activities. We hypothesized that seated computer users' spinal sagittal alignment could be continuously monitored using the HA and VD as explanatory variables. Then, we created a new experimental system that combined a 3D motion capture analysis and a radiographic measurement and we tested the accuracy of estimating the sagittal alignment while using a computer in the sitting position, with HA and VD as explanatory variables.

Data from 21 subjects were measured three times each, and each data set was analyzed individually. The findings revealed that HA and VD were strongly correlated with the sagittal alignment of the cervical, upper thoracic, and lumbar

**Table 5. Correlation coefficient (R) between sagittal spinal angles obtained by 3D-motion capture analysis and sagittal spinal parameters measured by whole-spine lateral X-ray.**

| Sagittal spinal parameter measured by Body surface landmark vs X-ray | Correlation coefficient (R) | Coefficient of determination (R²) |
|---|---|---|
| C2-C7 tilt angle vs COG-C7 SVA | 0.7259**** (0.6142 - 0.8091) | 0.5269 |
| C2-C7 tilt angle vs C2-C7 SVA | 0.7775**** (0.6846 - 0.8461) | 0.6046 |
| C2-C7 tilt angle vs CL | −0.1810<sup>NS</sup> (−0.3681 - 0.020) | 0.0327 |
| C7-T3 tilt angle vs T1 slope | 0.7700**** (0.6735 - 0.8407) | 0.5929 |
| C7-T3-T8 angle vs T1-T8 kyphosis | 0.5670**** (0.4136 - 0.6892) | 0.3215 |
| T3-T8-T12 angle vs T5-T12 kyphosis | 0.8304**** (0.7557 - 0.8838) | 0.6896 |
| T12-L3-S1 angle vs LL | 0.6185**** (0.4772 - 0.7286) | 0.3825 |

COG: center of gravity, SVA: sagittal vertical axis, CL: cervical lordosis, LL: lumbar lordosis.

Values in brackets indicate 95% confidence interval.

NS: not significant, *: $p < 0.05$, **: $p < 0.01$, ***: $p < 0.001$, ****: $p < 0.0001$.

spine. This means that by calculating approximate formulas for each individual, it is possible to estimate the sagittal alignment from the cervical to the lumbar spine using only HA and VD measurements. However, extending this estimation principle to a larger population by developing an estimation formulas for each individual is impractical. To identify the spinal alignment components that exhibited high correlations independent of gender, age, and body size, we combined data from all 21 subjects and verified the correlation between HA, VD, and sagittal alignment of each spinal region.

The results showed that the sagittal alignment of the cervical spine had a high correlation coefficient of over 0.8 with VD alone, and this correlation was even stronger when HA and VD were combined. In contrast, the correlation coefficients between HA/VD and the sagittal alignment of the thoracic and lower spine were less than 0.3. Objective variables indicating cervical alignment (C2-C7 tilt angle, C7-T3 tilt angle), increased when additional variables (gender, age, height, weight) were added to HA/VD. Adding age and gender as variables for the C2-C7 slope and C7-T3 slope, respectively, resulted in the highest correlation coefficients. Previous research has found that cervical flexion increases with age [13–15] and that the T1-slope, indicating upper thoracic anterior inclination, is significantly larger in males than in females [16]. Our results support these findings. Furthermore, C2-C7 tilt angle and C7-T3 tilt angles had a determination coefficient greater than 0.7 when using a quadratic approximation formula with HA/VD and greater than 0.8 when all additional explanatory variables were considered. These findings indicate that the cervical sagittal alignment can be estimated relatively accurately, regardless of gender, age, or body size, using the variables mentioned above.

However, this verification is based on measurements taken on the body surface, which raises concerns about the correlation with radiographic sagittal spinal parameters commonly used in spinal surgery. Therefore, standing whole-spine lateral radiographs were used to confirm the correlation between sagittal alignment measured using the same body surface landmarks as 3D motion analysis and radiographic sagittal parameters. As a result, the C2-C7 tilt angle had a high correlation coefficient above 0.7 with COG-C7 SVA and C2-C7 SVA, but no correlation with cervical lordosis. Although we were unable to locate a study that directly evaluated the relationship between C2-C7 SVA and cervical lordosis, Ikegami et al. reported that C2-C7 SVA and cervical lordosis do not always increase or decrease concurrently. Thus, it can be explained that the C2-C7 tilt angle, which correlates strongly with C2-C7 SVA, does not always correlate with cervical lordosis [15]. Furthermore, the C7-T3 tilt angle and T1 slope had a high correlation coefficient of 0.7700, demonstrating that the correlation obtained from the 3D motion analysis for estimating the cervical sagittal alignment is reliable. Moreover, the T3-T8-T12 angle and T5-T12 kyphosis had a high correlation coefficient of 0.8304 and a determination coefficient of 0.6896, indicating that the kyphosis of the mid-lower thoracic spine can be estimated with relatively high accuracy using body surface markers. Despite the promising correlation results for cervical alignment, there are limitations in applying these findings to the thoracic and lumbar regions where correlations were weaker.

In terms of the radiographic measurements, the reproducibility of most of the comparative parameters exceeded 0.6, except the T12-L3-S1 angle corresponding to lumbar lordosis, which had intra- and inter-examiner reproducibility less than 0.6, indicating the relative reliability of these measurements. Furthermore, the majority of intra and inter-examiner reliability of sagittal variable alignments used in routine radiographic measurements exceeded 0.8, indicating the radiographic measurements' overall reliability.

Many wearable devices warning against poor posture when attached to the head or neck have been sold, and their effectiveness is frequently exaggerated. Few studies have confirmed the accuracy and efficacy of these devices from a medical standpoint, and their credibility is limited. Luo et al. created a device that attaches to the cervical skin surface to monitor cervical alignment during daily activities, particularly during VDT work [17]. However, its relationship with sagittal spinal alignment measured on radiographic images has not been validated. Furthermore, its usefulness is limited by the need for prolonged skin contact, which may cause discomfort and potential skin problems.

Previous studies attempted to estimate the spinal sagittal alignment obtained from radiographic images using angular information derived from the body surface. A wearable device with inclinometers at the occiput, C7, L1, and S1 was

used to detect spinal inclination and estimate the cervical sagittal alignment by Kim et al. They showed a high correlation between sensor-derived estimates and radiographic measurements [18]. Similarly, Kawasaki et al. demonstrated that angles derived from the body surface—measured using the lateral canthus of the eye, tragus, and C7 spinous process in lateral photographs—correlate with cervical sagittal alignment on radiographic images [19]. This research, like our own, suggested that the segmental spinal angles obtained from the body surface are correlated with the cervical alignment measured on radiographs. However, while these reports show correlations between body surface-derived angular information and radiographic parameters, they have yet to achieve accurate sagittal alignment estimation using linear approximation.

Our method assesses the spinal sagittal alignment in the sitting position using only HA and VD. Therefore, if an AI-based tool capable of calculating HA and VD from facial images captured by built-in webcams on digital devices were created, laptop and smartphone users could assess their posture without the need for cumbersome wearable devices. Furthermore, such advancements may enable AI-powered posture monitoring systems to provide a convenient, noninvasive method for preventing posture-related musculoskeletal issues in modern office and school settings.

Our study has methodological and interpretational limitations. Ideally, spinal alignment should be measured using 3D motion analysis and fluoroscopic or radiographic imaging in the same subjects, but ethical considerations prevent healthy volunteers from being exposed to radiation for an extended period. We took an indirect approach, correlating HA/VD with sagittal alignment using body surface markers and radiographic parameters. Whole-spine radiographs from our bone metabolism clinic were used for X-ray validation because they are routinely taken for screening relatively healthy, middle-aged patients, mostly women. A seated whole-spine X-ray would be ideal, but it is uncommon in routine practice. Furthermore, the study's focus on seated computer work limits applicability, and more research is needed for smartphones and tablets. The small sample size and lack of randomization also reduced the results' generalizability to a larger population. Future research should suggest a larger, randomly selected sample with broader representation in terms of gender, age, height, and weight. Ideally, 3D motion analysis and X-rays should be conducted on the same subjects in identical postures to improve the acceptability and clinical relevance of the investigations.

In conclusion, this study showed that the sagittal alignment of the cervical spine can be calculated with relatively high accuracy during seated computer work using the variables of head angle and visual distance. Furthermore, including the variables of gender, age, height, and weight can enhance the estimation accuracy. These findings point to the potential for building devices and software that can monitor poor posture during VDT work, a growing social issue, in a convenient, continuous, and accurate manner, thereby contributing to the health maintenance of workers increasingly engaged in information terminal tasks

## Supporting information

**S1 Fig. Scatter plots of the correlation coefficient (a) and the coefficient of determination (b) between the head angle (HA)/visual distance (VD) and the objective parameters in a total of 63 (21 examinees x 3 trials) data sets from the 3D motion assessment, analyzed separately.** Bars indicate the 25th, 50th (median), and 75th percentiles. (TIFF)

**S2 Fig. Relationship between the spinal sagittal parameters obtained from body surface landmarks (x-axis) and their corresponding, actual spinal sagittal parameters commonly used in spine surgery (y-axis).** COG: center of gravity, SVA: sagittal vertical axis, deg: degree. (TIFF)

**S1 Table. Demographic data of the healthy volunteers for 3D motion capture analysis.** (DOCX)

**S2 Table. Demographic data of the patients for comparative analysis of sagittal spinal alignment between radiographic parameters and angles measured by body surface landmarks.**
(DOCX)

**S3 Table. Interrater correlation coefficients (ICC) of X-ray measurements.**
(DOCX)

**S4 Table. Coefficients of quadratic linear approximation (estimation formula) for C2-C7 tilt angle.**
(DOCX)

**S5 Table. Coefficients of quadratic linear approximation (estimation formula) for C7-T3 tilt angle.**
(DOCX)

## Acknowledgments

The authors acknowledge Mr. Yusei Takahashi of the Non-Profit Organization Natural Science for his valuable contribution to data acquisition for this study. The authors also thank the statistical expert, Mr. Hajime Yamakage (Satista Co., Ltd.), a statistical expert for reviewing and validating the statistical analysis methods and findings, and Enago (www.enago.jp) for the English language review.

## Author contributions

**Conceptualization:** Ko Hashimoto.

**Data curation:** Ko Hashimoto, Yusuke Sekiguchi, Kaho Matsuda, Masataka Hori, Yutaka Mizuno, Ryosuke Shibuya.

**Formal analysis:** Ko Hashimoto.

**Investigation:** Ko Hashimoto.

**Methodology:** Ko Hashimoto, Shin-Ichi Izumi.

**Supervision:** Shin-Ichi Izumi, Toshimi Aizawa.

**Validation:** Ko Hashimoto.

**Writing – original draft:** Ko Hashimoto.

**Writing – review & editing:** Ko Hashimoto, Kohei Takahashi, Takahiro Onoki, Kenichiro Yahata, Toshimi Aizawa.

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
