## [Decision Letter · Decision Letter 0]

PONE-D-24-55910High-Accuracy Spinal Alignment Monitoring Using Head Angle and Visual Distance in Computer UsersPLOS ONE

Dear Dr. Hashimoto,

Thank you for submitting your manuscript to PLOS ONE. After careful consideration, we feel that it has merit but does not fully meet PLOS ONE’s publication criteria as it currently stands. Therefore, we invite you to submit a revised version of the manuscript that addresses the points raised during the review process.

We look forward to receiving your revised manuscript.

Kind regards,

Holakoo Mohsenifar

Academic Editor

PLOS ONE

Journal Requirements:

Reviewers' comments:

Reviewer's Responses to Questions

**Comments to the Author**

1. Is the manuscript technically sound, and do the data support the conclusions?

Reviewer #1: Partly

Reviewer #2: Partly

2. Has the statistical analysis been performed appropriately and rigorously? 

Reviewer #1: Yes

Reviewer #2: Yes

3. Have the authors made all data underlying the findings in their manuscript fully available?

Reviewer #1: No

Reviewer #2: Yes

4. Is the manuscript presented in an intelligible fashion and written in standard English?

Reviewer #1: Yes

Reviewer #2: Yes

5. Review Comments to the Author

Reviewer #1: This study evaluates the reliability and validity of 3D motion analysis compared to traditional radiographic measurements for spinal alignment assessment, aiming to establish a radiation-free alternative for clinical practice. Twenty-one subjects underwent 3D analysis, while 32 underwent radiographic validation. Results showed strong correlations for cervical spine (R>0.8) but weaker for thoracic/lumbar regions. While the methodology appears sound, there are significant limitations in sample size and validation approach that need addressing. I have the following comments:

1. The most pressing concern is the insufficient sample size and lack of power analysis. The current sample sizes (n=21 and n=32) are not adequately justified, potentially leading to Type II errors. Please conduct a formal power analysis to determine appropriate sample sizes based on the primary outcome measures. Either justify the current numbers using appropriate statistical reasoning or consider increasing the study population to achieve adequate statistical power. Additionally, please clarify why different sample sizes were used for the two measurement methods and how this might affect the study's conclusions.

2. The Materials and Methods section requires substantial expansion regarding measurement methodology and analysis protocols. Please provide: (a) detailed description of the 3D motion analysis system setup, including camera positions, calibration procedures, and marker placement protocols; (b) specific measurement conditions and standardization procedures; (c) complete radiographic imaging protocols including positioning and measurement techniques; (d) reliability testing procedures including inter- and intra-rater reliability assessments; and (e) data quality control measures. This information is essential for ensuring study reproducibility and validity assessment.

3. The statistical analysis requires substantial revision. Please implement appropriate corrections for multiple comparisons (e.g., Bonferroni or False Discovery Rate corrections) given the multiple parameters being analyzed. Add cross-validation procedures and include effect sizes with 95% confidence intervals for all primary outcomes. Additionally, provide intraclass correlation coefficients (ICC) for reliability measures and Bland-Altman plots for agreement analysis between the two measurement methods. The correlation analysis should be supplemented with measures of absolute agreement and systematic bias assessment.

4. The data analysis methodology needs more detailed documentation. Please provide: (a) detailed description of all analysis scripts, including software versions and specific parameters used; (b) step-by-step data processing workflows; (c) criteria for data inclusion/exclusion; and (d) handling of missing or aberrant data points. Consider providing these analysis scripts as supplementary materials or in a public repository to enhance reproducibility. Additionally, please clarify the rationale for the chosen analysis methods and any assumptions made during the analysis process.

5. I suggest this manuscript requires significant improvement in language usage and structural consistency. Specific issues include: (a) inconsistent use of technical terminology throughout the paper; (b) unclear transitions between sections; (c) variable tense usage, particularly in the methods and results sections; and (d) complex sentence structures that obscure key findings. I strongly recommend engaging a professional English editing service to enhance the manuscript's readability and ensure clear communication of the research findings. Pay particular attention to standardizing terminology, maintaining consistent tense usage throughout each section, and improving overall flow and clarity.

These revisions are essential for improving the manuscript's scientific rigor and clarity. Please address each point thoroughly in your revision.

Reviewer #2: This study evaluates the reliability and validity of 3D motion analysis compared to traditional radiographic measurements for spinal alignment assessment, aiming to establish a radiation-free alternative for clinical practice. Twenty-one subjects underwent 3D analysis, while 32 underwent radiographic validation. Results showed strong correlations for cervical spine (R>0.8) but weaker for thoracic/lumbar regions. While the methodology appears sound, there are significant limitations in sample size and validation approach that need addressing. I have the following comments:

(1) The primary concern lies in the absence of a clear clinical relevance discussion. While the study demonstrates correlations between 3D motion analysis and radiographic measurements, it fails to establish the practical implications for clinical decision-making. The authors should elaborate on how the differences between measurement methods might impact clinical care, provide specific threshold values for clinically significant differences, and discuss the potential benefits and limitations of implementing 3D motion analysis in routine clinical practice. This discussion should include cost-effectiveness considerations and practical implementation challenges.

(2)A significant methodological issue is the lack of demographic and clinical characteristics analysis. The current manuscript does not adequately address how factors such as age, gender, body mass index, or pre-existing spinal conditions might influence measurement accuracy. I suggest that you should provide detailed participant characteristics, conduct subgroup analyses where appropriate, and discuss how these factors might affect the generalizability of their findings. Additionally, inclusion and exclusion criteria should be more clearly defined and justified.

(3)The third critical point concerns the absence of measurement error analysis and reliability testing over time. While single-time-point correlations are presented, the study lacks information about test-retest reliability, minimal detectable change, and standard error of measurement for both techniques. I suggest that you should conduct and report comprehensive reliability analyses, including intra-rater and inter-rater reliability for both measurement methods, and provide data on the temporal stability of measurements. This information is crucial for determining the clinical utility of the 3D motion analysis system.

(4)The results presentation requires substantial improvement in terms of data visualization and organization. The current figures and tables do not effectively communicate the relationship between measurement methods or the distribution of differences. I suggest that you should include scatter plots with identity lines for method comparisons, Bland-Altman plots showing limits of agreement, and clear visual representations of measurement variations across different spinal regions.

(5) Finally, the discussion section needs major revision to address study limitations and future research directions more comprehensively. The current discussion does not adequately contextualize the findings within the existing literature or address potential sources of systematic bias. I suggest that you should provide a more balanced interpretation of their results, including detailed comparison with similar studies, thorough analysis of methodological limitations, and specific recommendations for future research to address current gaps.

(6) Additionally, I suggest that you should discuss the potential impact of technological advances and how their findings might influence future development of spine assessment tools.

6. PLOS authors have the option to publish the peer review history of their article (what does this mean? ). If published, this will include your full peer review and any attached files.

**Do you want your identity to be public for this peer review?** For information about this choice, including consent withdrawal, please see our Privacy Policy .

Reviewer #1: No

Reviewer #2: No

---

## [Author Response · Author response to Decision Letter 1]

16 Mar 2025

We have carefully revised and modified the manuscript in response to the reviewers' comments, addressing each point individually. Below, we provide our detailed and sincere responses. We hope that these revisions have improved the manuscript to a level that is suitable for publication in PLoS One.

Reviewer #1: This study evaluates the reliability and validity of 3D motion analysis compared to traditional radiographic measurements for spinal alignment assessment, aiming to establish a radiation-free alternative for clinical practice. Twenty-one subjects underwent 3D analysis, while 32 underwent radiographic validation. Results showed strong correlations for cervical spine (R>0.8) but weaker for thoracic/lumbar regions. While the methodology appears sound, there are significant limitations in sample size and validation approach that need addressing. I have the following comments:

1. The most pressing concern is the insufficient sample size and lack of power analysis. The current sample sizes (n=21 and n=32) are not adequately justified, potentially leading to Type II errors. Please conduct a formal power analysis to determine appropriate sample sizes based on the primary outcome measures. Either justify the current numbers using appropriate statistical reasoning or consider increasing the study population to achieve adequate statistical power. Additionally, please clarify why different sample sizes were used for the two measurement methods and how this might affect the study's conclusions.

We recognize that the point you raised is highly important.

As stated in L104-109 and L114-120, the power analysis was appropriately conducted by a professional statistician. The results indicated that 10 cases were required for three-dimensional motion analysis, while 29 cases were needed for radiographic analysis. To emphasize that all power analyses were performed by a professional statistician, we have added the phrase: “A power analysis was conducted by a professional statistical expert.”

Based on the results of the above power analysis, we increased the sample sizes to 22 and 32, respectively, considering potential variability. Since the correlation analysis between three-dimensional motion analysis and radiographic measurements serves as an independent validation, it is not necessarily required to be conducted with identical sample sizes. Instead, we determined the sample sizes based on the criterion that the required statistical power was sufficiently exceeded.

2. The Materials and Methods section requires substantial expansion regarding measurement methodology and analysis protocols. Please provide: (a) detailed description of the 3D motion analysis system setup, including camera positions, calibration procedures, and marker placement protocols; (b) specific measurement conditions and standardization procedures; (c) complete radiographic imaging protocols including positioning and measurement techniques; (d) reliability testing procedures including inter- and intra-rater reliability assessments; and (e) data quality control measures. This information is essential for ensuring study reproducibility and validity assessment.

Thank you very much for indicating important issues.

(a) To clarify the 3D motion analysis system setup, the statement was amended as below: “The eight infrared cameras (Raptor-H, Raptor Photonics, Larne, Northern Ireland, UK) used for three-dimensional motion analysis were positioned at a height of 2.4 meters, evenly spaced in a circular arrangement surrounding the subject. Calibration was performed according to the protocol provided by the workstation (MAC 3D, Motion Analysis Corporation, Santa Rosa, CA). Specifically, during a series of validation experiments, a rod with two markers placed at 50 cm intervals was moved around various locations within the marker detection range. This process was repeated until a signal indicating sufficient marker detection accuracy was obtained.”

The marker placement method is described in detail in L122–131.

(b) The standardization method for the subject's posture and movements is described in detail in L131–137.

(c) To provide a more detailed description of the radiographic conditions, we have revised the statement as follows: "Lateral whole-spine radiographs were taken with the subject (patient) standing upright with both upper limbs naturally relaxed at their sides. A long-cassette film capable of capturing the entire spine in a single exposure was used, and the tube-to-film distance was set at 150 cm."

(d) The method for ICC testing is described in L232-237, and the results are provided in L289-299.

(e) To ensure the reliability of the measurements, we have provided a detailed explanation in our responses to points 3 and 4.

3. The statistical analysis requires substantial revision. Please implement appropriate corrections for multiple comparisons (e.g., Bonferroni or False Discovery Rate corrections) given the multiple parameters being analyzed. Add cross-validation procedures and include effect sizes with 95% confidence intervals for all primary outcomes. Additionally, provide intraclass correlation coefficients (ICC) for reliability measures and Bland-Altman plots for agreement analysis between the two measurement methods. The correlation analysis should be supplemented with measures of absolute agreement and systematic bias assessment.

Thank you very much for your critical comments regarding our statistical analysis methods. We have carefully discussed your points with a professional statistician as below.

First, in this study, each specific spinal region was analyzed independently, and the statistical analyses did not involve direct comparisons between variables. Therefore, we determined that multiple comparison corrections, such as Bonferroni correction, were not necessary.

Additionally, we sincerely apologize for any ambiguity in our methodological description that may have led to concerns regarding the homology of trials for each subject. In this experiment, to enhance the reliability of the correlation analysis, each subject performed similar movements three consecutive times. The aim was to obtain a greater number of datasets (21 subjects × 3 trials = 63 datasets). Importantly, the three trials did not need to be completely homologous, nor was it necessary to demonstrate statistical homology.

To clarify this point, we have revised the methodological description as follows:

“To improve the reliability of the subsequent correlation assessment, each subject repeated movements in the same manner three consecutive times without repositioning the body-mounted markers to generate a greater number of datasets.” This revision has been made in L146-149 for clarity.

4. The data analysis methodology needs more detailed documentation. Please provide: (a) detailed description of all analysis scripts, including software versions and specific parameters used; (b) step-by-step data processing workflows; (c) criteria for data inclusion/exclusion; and (d) handling of missing or aberrant data points. Consider providing these analysis scripts as supplementary materials or in a public repository to enhance reproducibility. Additionally, please clarify the rationale for the chosen analysis methods and any assumptions made during the analysis process.

Thank you very much for your valuable comments. In accordance with the reviewer's instructions, we have revised and expanded the description of the data analysis methods as follows (L151-168).

“From the 3D coordinate data obtained from the motion analysis, the sagittal components coordinates were extracted and used for the angle and distance analysis of the sagittal plane. To remove outliers from the continuous time-series data (collected at 0.01-second intervals), a moving average was calculated using a window of 100 consecutive data points (equivalent to 1 second), centered on each data point. Data points that deviated by more than 100% from the moving average were considered outliers and removed. This criterion was verified because, during slow and continuous human movements, it is unlikely that marker coordinates will change by more than 100% within a 1-s period. The investigations were conducted using a combination of Microsoft Excel software (Redmond, WA), GraphPad Prism9 software (Dotmatics, San Diego, CA) and the original C++-based program. The HA was defined as the angle formed by the horizontal line and the line segment connecting the two markers attached to the temple of the glasses. The VD was defined as the distance between the front-side marker the vane of the glasses’ vane and the midpoint of the two markers on the laptop computer. Based on the cleaned coordinate data, we calculated the inclination of the line segments connecting adjacent markers placed on the spinous processes relative to the vertical axis, as well as the angles formed between these adjacent segments.”

5. I suggest this manuscript requires significant improvement in language usage and structural consistency. Specific issues include: (a) inconsistent use of technical terminology throughout the paper; (b) unclear transitions between sections; (c) variable tense usage, particularly in the methods and results sections; and (d) complex sentence structures that obscure key findings. I strongly recommend engaging a professional English editing service to enhance the manuscript's readability and ensure clear communication of the research findings. Pay particular attention to standardizing terminology, maintaining consistent tense usage throughout each section, and improving overall flow and clarity.

Thank you very much for your valuable comments.

We acknowledge that there were inconsistencies in terminology and some unclear parts in the flow of the text. To address this, we have utilized a professional English editing service to refine and clarify the manuscript.

These revisions are essential for improving the manuscript's scientific rigor and clarity. Please address each point thoroughly in your revision.

Reviewer #2: This study evaluates the reliability and validity of 3D motion analysis compared to traditional radiographic measurements for spinal alignment assessment, aiming to establish a radiation-free alternative for clinical practice. Twenty-one subjects underwent 3D analysis, while 32 underwent radiographic validation. Results showed strong correlations for cervical spine (R>0.8) but weaker for thoracic/lumbar regions. While the methodology appears sound, there are significant limitations in sample size and validation approach that need addressing. I have the following comments:

(1) The primary concern lies in the absence of a clear clinical relevance discussion. While the study demonstrates correlations between 3D motion analysis and radiographic measurements, it fails to establish the practical implications for clinical decision-making. The authors should elaborate on how the differences between measurement methods might impact clinical care, provide specific threshold values for clinically significant differences, and discuss the potential benefits and limitations of implementing 3D motion analysis in routine clinical practice. This discussion should include cost-effectiveness considerations and practical implementation challenges.

Thank you very much for your important and insightful comments. We have provided a detailed discussion and expansion on the specific applications, use cases, and significance of this principle in L369-392. Due to the length of the text, we have not transcribed it here.

(2)A significant methodological issue is the lack of demographic and clinical characteristics analysis. The current manuscript does not adequately address how factors such as age, gender, body mass index, or pre-existing spinal conditions might influence measurement accuracy. I suggest that you should provide detailed participant characteristics, conduct subgroup analyses where appropriate, and discuss how these factors might affect the generalizability of their findings. Additionally, inclusion and exclusion criteria should be more clearly defined and justified.

Thank you very much for your valuable and appropriate comments.

In L104-109 and L114-120, we demonstrated through power analysis that the sample size was sufficient. However, we acknowledge that the number of subjects was not large enough to conduct reliable subgroup analyses based on age, sex, or BMI. For this reason, we chose to integrate all data points obtained from subjects with varying age, sex, and BMI to derive a linear approximation formula that comprehensively includes these factors.

Additionally, regarding the selection criteria for participants, we included only those without a history of spinal surgery. This information has been added to L103-104 in the main text.

The data on participants' age, sex, height, weight, and BMI are provided in Supplemental Table 1 and 2.

(3) The third critical point concerns the absence of measurement error analysis and reliability testing over time. While single-time-point correlations are presented, the study lacks information about test-retest reliability, minimal detectable change, and standard error of measurement for both techniques. I suggest that you should conduct and report comprehensive reliability analyses, including intra-rater and inter-rater reliability for both measurement methods, and provide data on the temporal stability of measurements.

This information is crucial for determining the clinical utility of the 3D motion analysis system.

(4)The results presentation requires substantial improvement in terms of data visualization and organization. The current figures and tables do not effectively communicate the relationship between measurement methods or the distribution of differences. I suggest that you should include scatter plots with identity lines for method comparisons, Bland-Altman plots showing limits of agreement, and clear visual representations of measurement variations across different spinal regions.

Since the comments in (3) and (4) partially overlap, we have provided a consolidated response below. We have carefully discussed your points with a professional statistician as below.

We sincerely apologize for any ambiguity in our methodological description that may have led to concerns regarding the homology of trials for each subject. In this experiment, to enhance the reliability of the correlation analysis, each subject performed similar movements three consecutive times. The aim was to obtain a greater number of datasets (21 subjects × 3 trials = 63 datasets). Importantly, the three trials did not need to be completely homologous, nor was it necessary to demonstrate statistical homology (Bland-Altman plot).

To clarify this point, we have revised the methodological description as follows:

“To improve the reliability of the subsequent correlation assessment, each subject repeated movements in the same manner three consecutive times without repositioning the body-mounted markers to generate a greater number of datasets.”　This revision has been made in L146-149 for clarity.

The methodology for the ICC analysis is described in L232-237, and the results are presented in L288-299.

Also, thank you very much for your important comments regarding the clarity of the figures and tables. In response to your suggestion, rather than simply listing numerical values in a table, we have visualized the correlation coefficients and determination coefficients between the incorporated variables and the spinal segment angles using graphs. This approach enhances the visual representation of the correlation strength and the accuracy of the approximation (Figure 4 and 5).

(5) Finally, the discussion section needs major revision to address study limitations and future research directions more comprehensively. The current discussion does not adequately contextualize the findings within the existing literature or address potential sources of systematic bias. I suggest that you should provide a more balanced interpretation of their results, including detailed comparison with similar studies, thorough analysis of methodological limitations, and specific recommendations for future research to address current gaps.

We have str

---

## [Decision Letter · Decision Letter 1]

High-Accuracy Spinal Alignment Monitoring Using the Head Angle and Visual Distance in Computer Users

PONE-D-24-55910R1

Dear Dr. Hashimoto,

We’re pleased to inform you that your manuscript has been judged scientifically suitable for publication and will be formally accepted for publication once it meets all outstanding technical requirements.

Kind regards,

Kentaro Yamada, M.D., Ph.D.

Academic Editor

PLOS ONE

Reviewers' comments:

Reviewer #1: The authors have revised their manuscript well according to the reviewers' comments.Thank you for the efforts.

Reviewer #2: The authors have revised their manuscript well based on the reviewers' suggestions. Thank you for your effort.

---

## [Editor Report · Acceptance letter]

PONE-D-24-55910R1

PLOS ONE

Dear Dr. Hashimoto,

I'm pleased to inform you that your manuscript has been deemed suitable for publication in PLOS ONE. Congratulations! Your manuscript is now being handed over to our production team.

Kind regards,

on behalf of

Dr. Kentaro Yamada

Academic Editor

PLOS ONE